# COMBINE-ICMH: A DUAL-ADAPTER CO-TUNING FRAMEWORK IN IMAGE COMPRESSION FOR MACHINE AND HUMAN VISION

## ABSTRACT

To reduce the high training overhead of models for Image Compression for Machine and Human Vision (ICMH), the paradigm of fine-tuning pre-trained models has gained increasing attention. Among these, lightweight adapter-based approaches have emerged as efficient solutions. However, we argue that this paradigm suffers from two critical, yet overlooked flaws. First, existing frequency-domain adapters lack adaptability, often suppressing high-frequency details crucial for machine tasks. Second, fine-tuning the transform module alone introduces a "transform-entropy mismatch," as the frozen entropy model cannot adapt to the altered latent distribution. To address these challenges, we propose Combine-ICMH, a novel framework that enables the synergistic co-optimization of both the transform and entropy models. Specifically, we design a Spatial-Wavelet Modulation Adapter (SWMA) to enhance frequency adaptability and introduce a Channel Modulation Adapter (CMA) to directly fine-tune the entropy model, resolving the mismatch. Extensive experiments demonstrate that our method consistently outperforms state-of-the-art approaches on various downstream tasks, including classification, detection, and segmentation, while maintaining comparable parameter efficiency.

## 1 INTRODUCTION

The surge of visual data from applications like autonomous driving demands efficient compression for analysis under limited bandwidth. While Image Compression for Machine (ICM) (Chen et al., 2019) optimizes for machine tasks, its severe degradation of visual quality makes the results unsuitable for human oversight. Therefore, Image Compression for Machine and Human Vision (ICMH) (Wang et al., 2021; Lin et al., 2023) has become a critical research topic.

Currently, the dominant paradigm for achieving ICMH has shifted from designing multi-task pipelines (Bai et al., 2022; Codevilla et al., 2021) to a tuning framework that adapts a pre-trained image codec optimized for human vision (Liu et al., 2023a; 2022). The former, though capable of parallel processing for both machine and human vision, suffers from high training overhead, redundant task-specific decoders, and degraded R-D performance, making large-scale deployment impractical (Li et al., 2024). In contrast, this fine-tuning paradigm is built upon a shared codec backbone, integrating lightweight, task-specific plug-and-play modules that add only a few trainable parameters. This design achieves dynamic switching between human and machine vision applications through a highly efficient, plug-and-play adaptation, all while crucially preserving the full reconstruction fidelity of the pre-trained model.

However, existing fine-tuning methods still exhibit significant limitations. Some works, such as ICMH-Net (Liu et al., 2023b) and TransTIC (Chen et al., 2023), employ masking or prompting (Lester et al., 2021; Li & Liang, 2021) for adaptation, but these techniques are tightly coupled with specific architectures (e.g., Transformers (Liu et al., 2021)) and incur substantial computational overhead. Another fine-tuning paradigm based on lightweight adapters, represented by Adapt-ICMH (Li et al., 2024), better balances complexity and performance by inserting lightweight adapters into the transform module.

Nevertheless, we identify two overlooked flaws in current adapter-based fine-tuning frameworks (Li et al., 2024; Zhao et al., 2025). First, they lack sufficient adaptability in the frequency domain. The Fourier-based frequency modulation adapter in Adapt-ICMH (Li et al., 2024) not only tends to suppress high-frequency components crucial for detection and segmentation, but also provides only limited enhancement for low-frequency components that carry critical global context for classification. The second, more critical flaw is the "transform–entropy mismatch". Current methods introduce adapters solely into the transform network (e.g., encoder or decoder), altering the latent representations. Meanwhile, the frozen entropy model is optimized for the original human-vision task and cannot accurately estimate the distribution of these altered latent representations. This misalignment leads to suboptimal bitrate estimation and degraded performance.

To address these critical challenges, we propose two targeted innovations. First, to combat insufficient frequency adaptability, we design the novel Spatial-Wavelet Modulation Adapter (SWMA). This adapter eliminates spatial redundancy through a dedicated spatial branch, while its parallel wavelet branch surpasses the limitations of Fourier-based methods by simultaneously enhancing crucial low-frequency components and preserving high-frequency ones. More critically, to resolve the "transform-entropy mismatch," we propose the Channel Modulation Adapter (CMA). By directly fine-tuning the entropy model, CMA realigns its probability estimates with the altered latent distribution, thereby addressing a core limitation of existing adapter-based methods.

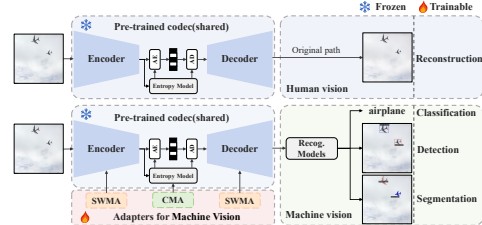

Figure 1: The framework of our Combine-ICMH. Both human and machine vision tasks share a pre-trained codec. By disabling the inserted adapters, the proposed framework could covert to base codec for human vision tasks.

Building upon these two innovations, we propose Combine-ICMH, an ICMH framework that co-optimizes the feature transformation and its entropy estimation. As shown in Figure 1, we insert SWMA into the base codec's transform network and integrate CMA into its entropy model. This synergistic fine-tuning strategy effectively addresses the limitations of existing methods. Our main contributions are as follows:

- We propose the Spatial-Wavelet Modulation Adapter (SWMA), which addresses the insufficient frequency adaptability in prior work. It processes spatial and frequency information in parallel, leading to superior fine-tuning performance.

- We design the lightweight Channel Modulation Adapter (CMA), which resolves the "transform-entropy mismatch" in existing frameworks by aligning the entropy model with the shifted latent feature distribution.

- Based on the proposed adapters, we develop Combine-ICMH, an ICMH framework that synergistically fine-tunes both the transform and entropy models. Experimental results demonstrate that our method consistently outperforms existing ICMH frameworks across various machine vision tasks while maintaining a comparable level of complexity.

## 2 RELATED WORKS

### 2.1 LEARNED IMAGE COMPRESSION

Learned image compression (LIC) has advanced significantly since Ballé et al. (2017) introduced the first end-to-end image compression framework. Nowadays, state-of-the-art (SOTA) LIC models outperform the best traditional codecs, such as VVC (Bross et al., 2021) and HEVC (Sullivan et al., 2012). Progress in this field has primarily focused on two core components: the transform module and the entropy model.

In the domain of transform module, Cheng et al. (2019) improved compression by stacking residual blocks, while Ghorbel et al. (2023) employed ConvNeXt modules to balance performance and complexity. In addition, attention- and Transformer-based methods have been explored: Zou et al. (2022) introduced window-based local attention, and Zhu et al. (2022) built a pure Swin Transformer architecture.

In parallel, improving the entropy model is essential for efficient coding. For instance, Ballé et al. (2018) introduced the hyperprior network, leveraging side information to capture spatial correlations. Building on this, Minnen et al. (2018) proposed an autoregressive model. To further enhance coding efficiency, He et al. (2021) and Jiang & Wang (2023) fused spatial–channel and multi-reference contexts, respectively.

However, LIC models, optimized for human vision, are suboptimal for machine tasks. Existing fine-tuning approaches generally adjust only the transform module, overlooking its tight coupling with the entropy model. This oversight causes a "transform-entropy mismatch," which results in inefficient rate estimation and degraded performance.

## 2.2 IMAGE COMPRESSION FOR MACHINE AND HUMAN VISION

To address the conflict between machine vision (Ding et al., 2022; Sun et al., 2019; Xie et al., 2021a) and human perception in image coding, Image Compression for Machine and Human Vision (ICMH) has emerged.

The primary objective of ICMH is to reconcile human perception with machine analysis. For instance, Choi & Bajić (2022) divided the latent representation into a "base layer" and an "enhancement layer," which are transmitted via a scalable bitstream to decouple machine vision features from human vision information. Another work (Liu et al., 2022) introduced a framework that performs machine analysis directly on compressed representations. While such designs improve perceptual quality over earlier ICM approaches (Chamain et al., 2021; Feng et al., 2022; Le et al., 2021), they suffer from substantial training and storage overhead, limiting their practical utility (Li et al., 2024).

Recently, fine-tuning frameworks that adapt pre-trained image codecs (Fischer et al., 2022; Chen et al., 2023) have gained more attention. Notably, Chen et al. (2023) proposed a prompt-based tuning framework. These approaches leverage pre-trained models to reduce training overhead without compromising the human perceptual quality of the pretrained model. However, such methods are often restricted to specific architectures and suffer from low fine-tuning efficiency. Furthermore, Li et al. (2024) extended their applicability, proposing a fine-tuning paradigm based on lightweight adapters.

# 3 PROPOSED METHOD

## 3.1 EMPIRICAL ANALYSIS AND MOTIVATIONS

To efficiently adapt human-oriented LIC models to machine vision tasks without compromising R–D performance of the pretrained models, recent works (Li et al., 2024; Zhao et al., 2025) have adopted fine-tuning paradigms based on lightweight adapters. The core idea of this paradigm involves fine-tuning lightweight, plug-and-play adapters while freezing the base codec. Although this paradigm is promising, we identify two key limitations in current adapter designs.

First, we observe a critical limitation in the frequency adaptability of existing frequency modulation adapters. Different machine vision tasks exhibit distinct preferences for frequency components: fine-grained tasks like instance segmentation rely on mid- and high-frequency components, whereas coarse-grained tasks like image classification focus more on low-frequency components. An ideal adapter should flexibly cater to these diverse demands. However, as shown in Fig. 2, existing Fourier-based adapters excessively attenuate high-frequency components while failing to sufficiently amplify low-frequency ones. This lack of frequency adaptivity exposes the shortcomings of current frequency-modulation strategies. To address these issues, we propose

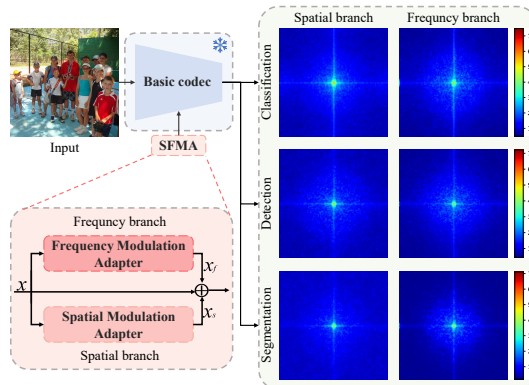

Figure 2: Power spectral density (PSD) maps of intermediate features from the Li et al. (2024) for three downstream tasks.

a Spatial–Wavelet Modulation Adapter (SWMA) that leverages the multi-level wavelet transform to deliver more precise and effective feature modulation.

Beyond frequency adaptivity, we identify a crucial architectural issue in existing adapter-based fine-tuning frameworks: the "transform-entropy mismatch". Learned image compression (LIC) comprises two key components: nonlinear transform module and entropy model. Previous methods (Chen et al., 2023; Li et al., 2024) typically fine-tune only one component, usually the transform module, which introduces a substantial side effect. Specifically, fine-tuning the transform alone alters the latent distribution. Consequently, the entropy model, which was originally optimized for human vision, becomes unsuitable for the new latent representation tailored for machine vision. This mismatch results in inefficient bitrate estimation (e.g., increased bitrate) and degraded performance.

This inefficiency can be formally analyzed. In the hyperprior entropy model (Ballé et al., 2018), the rate of latent representation $R(\hat{y})$ is determined by the estimated conditional entropy. When the learned entropy model mismatches the actual distribution of the representation (i.e., the hyperprior $\hat{z}$ deviates from the ideal $\tilde{z}$ ), the actual rate can be decomposed into the ideal rate and a penalty term, $\Delta bpp$. This decomposition is given by (see Appendix E for detailed derivation):

$$R(\hat{y}) = \mathbb{E}\left[-\log_2(p_{\hat{y}|\hat{z}}(\hat{y} \mid \hat{z}))\right] = \mathbb{E}\left[-\log_2(p_{\hat{y}|\tilde{z}}(\hat{y} \mid \tilde{z}))\right] + \Delta bpp \tag{1}$$

To validate our mismatch hypothesis and demonstrate the performance penalty it incurs, we conducted a staged freezing experiment. This experiment employed the TIC architecture (Lu et al., 2021), where we selectively fine-tuned layers of the 8-layer analysis/synthesis transforms ($g_a, g_s$) and the 4-layer hyperprior encoder/decoder ($h_a, h_s$). Table 1 specifies the indices of the unfrozen (i.e., trainable) layers for each configuration. Performance was evaluated using bits per pixel (bpp), mAP, and the latent correlation $\rho$ as defined in Zhu et al. (2022).

Table 1: Analysis of a stage-wise freezing experiment on the TIC (Lu et al., 2021) for object detection on COCO (Lin et al., 2014). In this experiment, specific network components were frozen to show the effects of fine-tuning only the transform modules ($g_a, g_s$) or fine-tuning them along with the entropy models ($h_a, h_s$).

| $g_a$ | $g_s$ | $h_a$ | $h_s$ | $bpp$ | $mAP$ | $\rho$ | Param(M) |
|-------|-------|-------|-------|-------|-------|--------|----------|
| 0-7   | 0-7   | 0-3   | 0-3   | 0.0639 | 37.63 | 0.239 | 7.51 |
| 1,3,5 | 2,4,6 | \     | \     | 0.0829 | 37.21 | 0.289 | 3.22 |
| 0-7   | 0-7   | \     | \     | 0.0881 | 37.32 | 0.314 | 5.48 |
| 1,3,5 | 2,4,6 | 0-3   | 0-3   | 0.0685 | 37.42 | 0.260 | 5.25 |

As highlighted by rows 3 and 4 of Table 1, forgoing entropy model fine-tuning incurs a critical penalty. Despite employing more parameters, fine-tuning only the transform module simultaneously reduces task accuracy and compression efficiency (i.e., increases bitrate). We attribute this counter-intuitive result to the "transform-entropy mismatch," where the frozen entropy model fails to adapt to the distribution of the latent representation in the new domain, leading to significant coding redundancy. The sharp increase in latent correlation $\rho$ of the normalized latents further corroborates this conclusion.

To resolve this mismatch, we introduce the Channel Modulation Adapter (CMA), which modulates the entropy model via channel-wise adaptation to match shifted latent distributions. Finally, we combine our two specialized modules, the SWMA and the CMA, into a unified framework. This framework co-optimizes the transform and entropy modules, mitigating performance bottlenecks caused by distribution shifts and enabling efficient adaptation.

## 3.2 FRAMEWORK

To address the core problem of the mismatch between the transform and entropy models in existing adapter-based fine-tuning, we propose Combine-ICMH, a new framework that synergistically fine-tunes both modules through two efficient, plug-and-play adapters.

As illustrated in Figure 3, we insert SWMA into the transform modules to adapt them for downstream machine vision tasks. Unlike previous Fourier-based adapters, SWMA leverages a multi-level wavelet transform to enable level-by-level processing of frequency components. SWMA thus

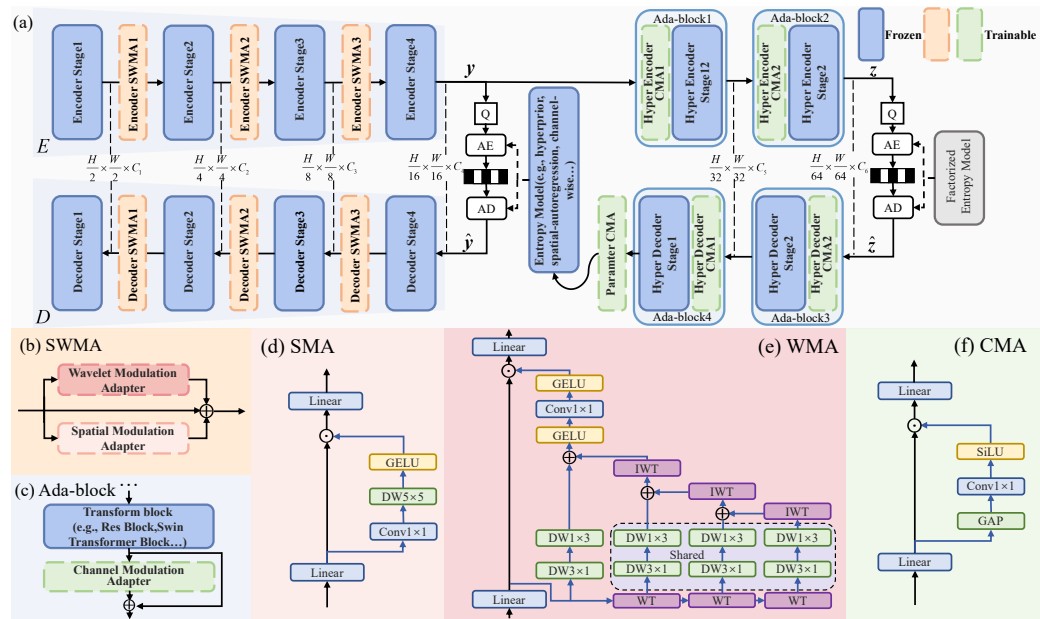

Figure 3: Overall architecture of Combine-ICMH. (a) Main pipeline. SWMA is integrated into the encoder ($E$) and decoder ($D$) of the base codec for feature adaptation, while CMA fine-tunes features from the hyperprior network to mitigate transform–entropy mismatch. (b) The proposed SWMA, comprising an SMA (d) and a WMA (e). (c) The structure of the Ada-block. (d) Architecture of the SMA. (e) Architecture of the WMA. (f) Architecture of the CMA.

enables a task-aware modulation of the frequency spectrum, selectively preserving high-frequency components (i.e., edges and boundaries) for precise localization in detection and segmentation, while enhancing low-frequency ones that capture global context for classification.

However, fine-tuning only the transform modules typically leads to the "transform-entropy mismatch". To tackle this challenge, we further integrate the CMA into the hyperprior network ($h_a$, $h_s$). This adapter adjusts the entropy model via channel-wise modulation of the hyperprior, enabling it to precisely align with the latent distribution after fine-tuning.

In line with prior works (Chen et al., 2023; Li et al., 2024; Park et al., 2025), our strategy involves freezing the pre-trained codec and optimizing only the adapters, which significantly reduces training overhead while preserving the codec's performance for human vision. Crucially, these adapters can be disabled at inference time, allowing our framework to losslessly revert to the base codec and maintain identical R-D performance to the baseline codec.

For a fair comparison with prior works, we adopt the loss function from (Chen et al., 2023):

$$\mathcal{L} = \mathcal{R} + \lambda \cdot \mathcal{D}_{task} \tag{2}$$

where $\mathcal{R}$ denotes the bitrates estimated by the entropy model, $\mathcal{D}_{task}$ is the task-specific perceptual distortion (detailed in Appendix A), and $\lambda$ is the trade-off hyper-parameter.

### 3.3 SPATIAL-WAVELET MODULATION ADAPTER(SWMA)

The proposed Spatial-Wavelet Modulation Adapter (SWMA) adopts a parallel dual-branch structure to efficiently refine features. Its Spatial Modulation Adapter (SMA) branch removes spatial redundancy, while the Wavelet Modulation Adapter (WMA) branch modulates frequency components.

**Spatial Modulation Adapter (SMA).** The SMA, shown in Figure 3 (d), is designed to reduce spatial redundancy. To achieve this, an input feature $x$ is first projected into the lower-dimensional feature $x_s$ via a linear layer $W_{\text{down}}^s$. A spatial map ($x_{smap}$) is then generated from $x_s$ through a $1 \times 1$ convolution and a $5 \times 5$ depth-wise convolution for a larger receptive field. Finally, the feature $x_s$ is

modulated by the activated spatial map, and the result is up-projected to produce the output $x_{sout}$. The process is formulated as:

$$x_s = W_{down}^s(x) \tag{3}$$

$$x_{smap} = \text{DWConv5} \times 5(\text{Conv1} \times 1(x_s)) \tag{4}$$

$$x_{sout} = W_{up}^s(x_s \odot \text{GELU}(x_{smap})) \tag{5}$$

where $W_{down}^s$ and $W_{up}^s$ are linear projection layers, GELU denotes the Gaussian Error Linear Unit activation (Hendrycks & Gimpel, 2016) and $\odot$ represents element-wise multiplication.

**Wavelet Modulation Adapter (WMA).** Our WMA, detailed in Fig. 3 (e), is designed to hierarchically process frequency-domain features. It operates within a bottleneck, where the intermediate feature $x_w$ undergoes a multi-level Haar wavelet decomposition. The decomposed sub-bands are passed through parameter-shared $3\times1$ and $1\times3$ depth-wise convolutions that efficiently capture directional patterns. As depicted in Figure 3 (e), these processed sub-bands are then progressively fused through a three-level inverse wavelet transform. The reconstructed output is processed by a $1 \times 1$ convolution to generate a map $x_{wmap}$ that modulates the intermediate feature $x_w$. Finally, the modulated feature is projected back to the higher dimension. The process can be formulated as:

$$x_w = W_{down}^w(x) \tag{6}$$

$$x_{wout} = W_{up}^w(x_w \odot x_{wmap}) \tag{7}$$

where $W_{down}^w$ and $W_{up}^w$ are linear projection layers, $\odot$ represents element-wise multiplication, and $x_{wmap}$ denotes the wavelet processing branch detailed in Figure 3 (e), and the full formulation of $x_{wmap}$ is provided in the Appendix B.

### 3.4 CHANNEL MODULATION ADAPTER (CMA)

Our proposed CMA module resolves the mismatch between the entropy model and the transform module by modulating channel features within the hyperprior network. Specifically, it first down-projects the input feature $x$ and then generates a channel attention vector $x_{cmap}$ using global average pooling, a $1\times1$ convolution, and a SiLU activation (Elfwing et al., 2018). The final output is then obtained by up-projecting the down-projected feature $x_c$ after being modulated by $x_{cmap}$. The process is formulated as:

$$x_c = W_{down}^c(x) \tag{8}$$

$$x_{cmap} = \text{SiLU}(\text{Conv1x1}(\text{GAP}(x_c))) \tag{9}$$

$$x_{cout} = W_{up}^c(x_{cmap} \odot x_c) \tag{10}$$

where $W_{down}^c$ and $W_{up}^c$ are the down/up-projection layers (with a ratio of $1/8$) and $\odot$ represents element-wise multiplication.

## 4 EXPERIMENTS

**Implementation Details.** Following Chen et al. (2023), we adopt a simplified version of the TIC (Lu et al., 2021) as our base codec. The proposed SWMA, which uses an intermediate bottleneck dimension of 64, is integrated into the first three stages of the transform module, adhering to the setup in Li et al. (2024). The CMA, with an intermediate dimension of 16, is inserted into the hyperprior network. As detailed in Figure 3 (c), four of these CMA modules are placed within the hyperprior network, and the final one is placed after it to ensure alignment with the latent representation.

**Training Details and Datasets.** We evaluate our framework on three downstream machine vision tasks: classification, object detection, and instance segmentation. For training, we use the ImageNet-train dataset (Deng et al., 2009) and the COCO2017-train dataset (Lin et al., 2014), aligning with prior works (Chen et al., 2023).

We adopt a parameter-efficient fine-tuning strategy, optimizing only the proposed adapters while keeping the base codec frozen. The training is guided by a task-aware loss (Eq. 2) computed using pre-trained downstream models: ResNet-50 (He et al., 2016) for classification, Faster R-CNN (Ren et al., 2015) for object detection, and Mask R-CNN (He et al., 2017) for instance segmentation. For data preparation, all images are randomly cropped to $256\times256$. The classification adapters

are trained for 8 epochs with a batch size of 16, while the detection and segmentation adapters are trained for 40 epochs with a batch size of 8. All experiments are implemented in PyTorch and run on a single NVIDIA RTX 4090 with Adam (learning rate is 1e-4). Further details are in the appendix.

**Evaluation Metrics.** We evaluate the compression rate in bits per pixel (bpp) alongside task-specific performance. Specifically, we report top-1 accuracy on the ImageNet-val dataset for classification (using a pre-trained ResNet-50), and mAP on the COCO2017-val dataset for object detection and instance segmentation (using models from the Detectron2 library (Wu et al., 2019)). Rate-performance curves are generated by training models at different $\lambda$ values. To summarize these curves, we report two BD metrics: BD-Rate and BD-acc/mAP.

## 4.1 MAIN RESULTS

To demonstrate the effectiveness of the proposed method, we compare it with five SOTA approaches: Channel Selection (Liu et al., 2022), ICMH-Net (Liu et al., 2023b), TransTIC (Chen et al., 2023), Adapt-ICMH (Li et al., 2024), and SVD-LoRA (Park et al., 2025). The evaluation is conducted on three standard machine vision tasks. For a comprehensive comparison, we adopt two established metrics: the Bjontegaard Delta Rate (BD-rate) (Bjontegaard, 2001) to measure bitrate savings (lower is better), and BD-acc/mAP (Chen et al., 2023) to quantify the average performance gain at equivalent bitrates (higher is better). Figure 4 visualizes the rate-performance curves, while Table 2 provides a detailed quantitative summary.

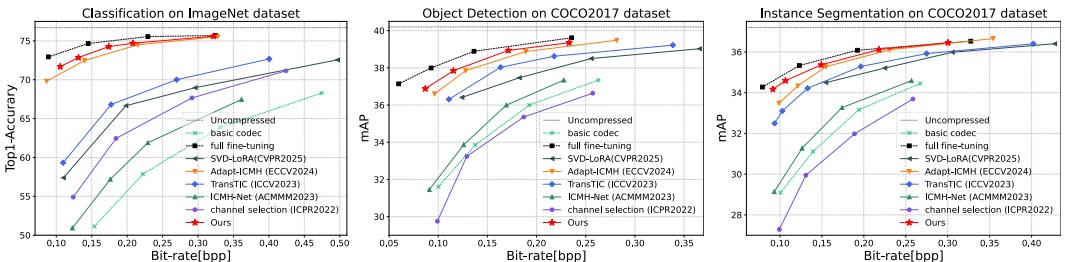

Figure 4: Rate-Accuracy performance comparison under different machine vision tasks.

Figure 4 and Table 2 show that our method consistently outperforms competing approaches across the three tasks. It maintains a trainable parameter count comparable to the SOTA method (3.8 % vs. 4.2 %) while reducing computational complexity (kMACs), thus achieving a superior performance–efficiency trade-off. To further demonstrate SWMA's efficiency, we evaluate a lightweight variant that excludes CMA. As Table 2 indicates, despite having fewer trainable parameters and lower computational cost, this variant still reaches SOTA performance, underscoring SWMA's exceptional effectiveness and frequency-adaptation ability.

Table 2: Performance comparison of different methods on three machine tasks, using TIC as the base codec. We report the number of trainable parameters, computational complexity, and two BD metrics (Bjontegaard, 2001): BD-rate and BD-acc/mAP.

| Method | Classification | | Detection | | Segmentation | | kMACs | Trainable |
| | BD-rate↓ | BD-acc↑ | BD-rate↓ | BD-mAP↑ | BD-rate↓ | BD-mAP↑ | (/pixel) | Params ↓(M) |
|---|---|---|---|---|---|---|---|---|
| full fine-tuning | / | 17.67 | -75.03% | 4.56 | -67.95% | 3.76 | 331.0 | 7.511(100%) |
| channel selection | -37.18% | 6.28 | 6.24% | -0.50 | 16.51% | -0.95 | **167.7** | 0.915(12.2%) |
| ICMH-Net | -18.76% | 3.36 | -9.82% | 0.67 | -10.94% | 0.66 | 364.2 | 3.982(53.0%) |
| TransTIC | -58.53% | 9.96 | -47.50% | 2.83 | -46.48% | 2.69 | 534.6 | 1.620(21.6%) |
| SVD-LoRA | -58.39% | 7.83 | -34.87% | 1.96 | -43.43% | 1.69 | 331.0 | **0.093(1.2%)** |
| Adapt-ICMH | -87.06% | 16.68 | -56.22% | 3.60 | -52.33% | 3.21 | 360.4 | 0.287(3.8%) |
| Ours(w/o CMA) | -87.45% | 16.85 | -60.90% | 3.85 | -56.60% | 3.36 | 359.6 | 0.276(3.7%) |
| Ours | **-91.38%** | **17.15** | **-61.68%** | **4.10** | **-62.20%** | **3.49** | 359.9 | 0.319(4.2%) |

## 4.2 ABLATION STUDY

**Effectiveness of the SWMA.** To evaluate the contributions of our SMA and WMA, we conduct a component-wise ablation study with their counterparts from SFMA (Li et al., 2024) (Table 3). The results show the superiority of our proposed modules. First, our SMA consistently improves performance with fewer parameters (e.g., row 2 vs. 4 and row 1 vs. 3). Second, our WMA yields a substantial performance gain over its counterpart from SFMA (Li et al., 2024) (e.g., row 1 vs. 4 and row 2 vs. 3). By integrating these two components, our SWMA achieves SOTA performance with lower trainable parameters.

Table 3: Ablations on different variants of SWMA. We replaced the spatial and frequency adapters from Li et al. (2024) with our proposed adapters, achieving a significant performance improvement.

| FMA | WMA (ours) | SMA | SMA (ours) | Classification | | Detection | | Segmentation | | Params |
|---|---|---|---|---|---|---|---|---|---|---|
| | | | | BD-rate↓ | BD-acc↑ | BD-rate↓ | BD-mAP↑ | BD-rate↓ | BD-mAP↑ | ↓(M) |
| ✓ | | ✓ | | -87.06% | 16.68 | -56.22% | 3.60 | -52.33% | 3.21 | 0.287(3.8%) |
| | ✓ | | ✓ | **-87.45%** | **16.85** | **-60.90%** | **3.85** | **-56.60%** | **3.36** | 0.276(3.7%) |
| ✓ | | | ✓ | -87.12% | 16.81 | -56.94% | 3.73 | -54.45% | 3.23 | 0.263(3.5%) |
| | ✓ | ✓ | | -87.38% | **16.85** | -58.48% | 3.77 | -56.47% | 3.31 | 0.300(4.0%) |

**Effectiveness of the CMA.** To verify the efficacy of the proposed CMA, we conduct experiments based on SFMA and our proposed SWMA, with results presented in Table 4. The findings reveal two key points. First, the CMA significantly boosts the performance of both SFMA and SWMA with minimal parameter overhead, highlighting the effectiveness of our channel modulation approach in addressing the "transform–entropy mismatch". Second, applying the CMA alone to the basic codec yields negligible improvement (or even worse). This is expected because the CMA is designed as a corrective module to mitigate distribution drift from fine-tuning, not as a general feature enhancer.

Table 4: Ablations on CMA.

| SFMA | SWMA | CMA | Classification | | Detection | | Segmentation | | Params |
|---|---|---|---|---|---|---|---|---|---|
| | | | BD-rate↓ | BD-acc↑ | BD-rate↓ | BD-mAP↑ | BD-rate↓ | BD-mAP↑ | ↓(M) |
| | ✓ | | -87.45% | 16.85 | -60.90% | 3.85 | -56.60% | 3.36 | 0.276(3.7%) |
| ✓ | | | -87.06% | 16.68 | -56.22% | 3.60 | -52.33% | 3.21 | 0.287(3.8%) |
| ✓ | | ✓ | -87.67% | 17.04 | -57.46% | 3.91 | -56.97% | 3.26 | 0.331(4.4%) |
| | ✓ | ✓ | **-91.38%** | **17.15** | **-61.68%** | **4.10** | **-62.20%** | **3.49** | 0.319(4.2%) |
| | | ✓ | 0.37% | -0.06 | -1.88% | 0.11 | -1.49% | 0.08 | 0.044(0.6%) |

**Effect on the middle dimension.** Next, we analyze the middle dimension of SWMA (see Table 5). The results show a significant performance improvement as the dimension increases from 32 to 64. However, further increases beyond 64 yield diminishing gains while the parameter count keeps growing. This indicates that a dimension of 64 offers the best trade-off between performance and efficiency, so we adopt it as our final configuration.

Table 5: Ablations on the middle dimension of SWMA.

| Middle Dimension | Classification | | Detection | | Segmentation | | Params |
|---|---|---|---|---|---|---|---|
| | BD-rate↓ | BD-acc↑ | BD-rate↓ | BD-mAP↑ | BD-rate↓ | BD-mAP↑ | ↓(M) |
| 32 | -83.10% | 16.48 | -57.43% | 3.75 | -58.11% | 3.33 | 0.170(2.3%) |
| 64 | -91.38% | 17.15 | -61.68% | 4.10 | -62.20% | 3.49 | 0.319(4.2%) |
| 128 | -92.09% | 17.80 | -64.67% | 4.25 | -65.49% | 3.58 | 0.692(9.2%) |

## 4.3 APPLICATION ON LARGER SOTA METHOD

To further validate the generalization of our framework, we apply it to another SOTA codec, DCAE (CVPR'25) (Lu et al., 2025). DCAE presents a particularly challenging baseline due to its significantly larger parameter count (119M) and a more complex, dictionary-based entropy model. We

evaluate our method with competing ICMH methods Chen et al. (2023); Li et al. (2024) on the object detection task. The results in Table 6 show that our method outperforms competing methods while maintaining a comparable parameter count. These experiments confirm the strong generalization ability of our framework, showcasing its robustness across diverse backbones and its adaptability to different entropy models.

Table 6: Performance comparison for object detection on the COCO2017 dataset using the challenging DCAE model.

| Method | Detection | | Trainable |
| | BD-rate↓ | BD-mAP↑ | Params ↓(M) |
|---|---|---|---|
| full fine-tuning | -84.68% | 4.00 | 119.760 (100%) |
| TransTIC | -49.27% | 2.34 | 1.128 (0.94%) |
| Adapt-ICMH | -64.61% | 2.86 | 0.359 (0.30%) |
| Ours | **-67.87%** | **3.43** | 0.434 (0.36%) |

### 4.4 QUALITATIVE RESULTS

Figure 5 provides visual comparisons that underscore the superiority of our method. Even at lower bitrates, our approach captures fine-grained details, such as the person's tie, that are missed by competing methods. A direct comparison with Adapt-ICMH further shows that our method not only preserves critical high-frequency information (e.g., edges and textures) but also effectively enhances essential low-frequency components. We attribute these advantages to the frequency-adaptation capability of our SWMA module.

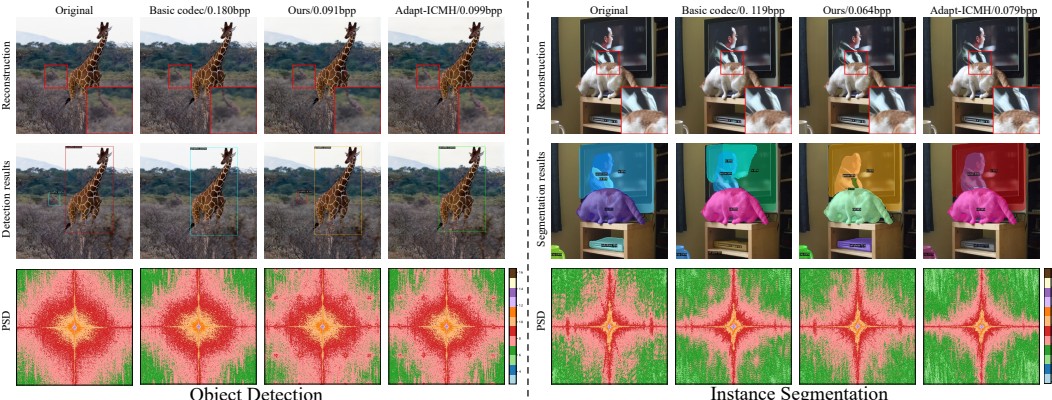

Figure 5: Qualitative comparison of different methods on detection and segmentation. The figure displays (from top to bottom): the original and decoded images, their corresponding results, and the log Power Spectral Density (PSD) of these images.

## 5 CONCLUSION

In this paper, we address the limitations of existing adapter-based tuning frameworks by proposing Combine-ICMH, a novel framework for the synergistic co-tuning of both transform and entropy models. Specifically, the proposed Spatial-Wavelet Modulation Adapter (SWMA) performs efficient feature adaptation in both spatial and frequency domains to meet the demands of downstream machine tasks. Concurrently, our proposed Channel Modulation Adapter (CMA) directly fine-tunes the entropy model, resolving the critical mismatch between the adapted features and the pre-trained entropy model. Experiments show that our method outperforms existing ICMH approaches across multiple machine vision tasks, while maintaining a comparable or even fewer number of trainable parameters. Additional experiments on more advanced backbones further demonstrate the robustness of our approach.

## 6 REPRODUCIBILITY STATEMENT

We have taken several steps to facilitate the reproduction of our results.

- **Datasets.** Section 4 details the dataset selection process and the complete pre-processing pipeline.
- **Hyper-parameters.** Appendix C enumerates every hyper-parameter used in the main experiments.
- **Compute.** Appendix C also summarizes the software and hardware specifications.
- **Evaluation.** Appendix A describes the task-specific perceptual-distortion losses.

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

# Appendix

## A  TASK-SPECIFIC PERCEPTUAL LOSS

To effectively guide the model's training, we employ a task-specific perceptual loss ($D_{task}$) (following Chen et al. (2023)). This perceptual loss is defined by calculating the Mean Squared Error (MSE) between features of the original image $x$ and the reconstructed image $\hat{x}$ (generated by image codecs), where the features are extracted from a pre-trained, task-relevant network.

For different downstream tasks, we utilize specific network feature layers and corresponding loss formulations. The detailed configurations are summarized in Table 7.

To visually illustrate the feature layers used for loss computation, we highlight their positions within the network architecture in Figure 6.

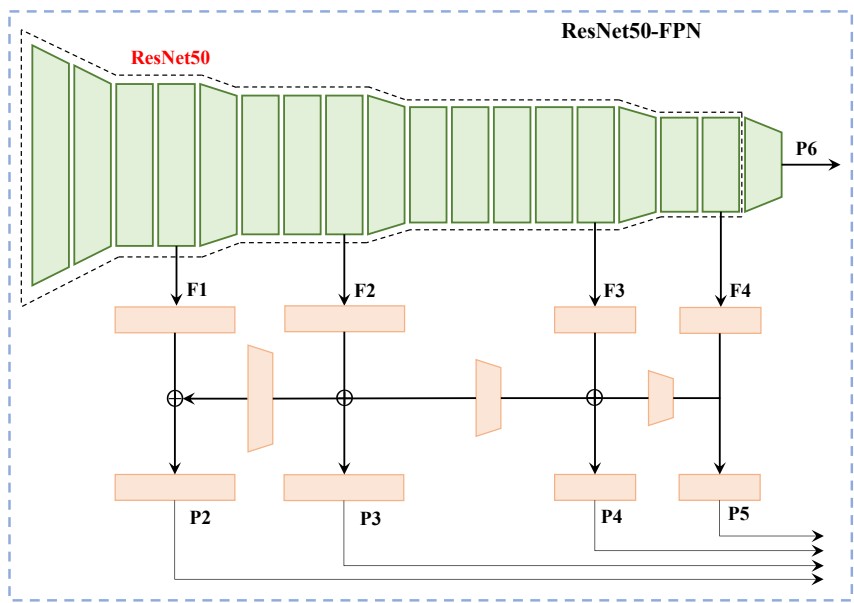

Figure 6: Network architecture of ResNet50-FPN.

## B  DETAIL INFORMATION FOR WMA

This section provides supplementary details on the calculation formula for the wavelet modulation map $w_{map}$.

Table 7: Perceptual Loss Configurations for Different Downstream Tasks.

| Downstream Task | Network | Feature Layers Used | Loss Formulation |
|---|---|---|---|
| Classification | ResNet-50 (He et al., 2016) | $F_1, F_2, F_3, F_4$ | $D(x, \hat{x}, G) = \frac{1}{4} \sum_{j=1}^{4} \mathrm{MSE}(F_j(x), F_j(\hat{x}))$ |
| Object Detection | Faster R-CNN (Ren et al., 2015) | $P_2, P_3, P_4, P_5, P_6$ | $D(x, \hat{x}, G) = \frac{1}{5} \sum_{j=2}^{6} \mathrm{MSE}(P_j(x), P_j(\hat{x}))$ |
| Instance Segmentation | Mask R-CNN (He et al., 2017) | $P_2, P_3, P_4, P_5, P_6$ | |

In the computation, the intermediate feature map $w_m$, along with the output of each wavelet transform level, is processed by a sequence of 3x1 and 1x3 depthwise convolutions. The process is as follows:

$$\mathrm{DW}(T) = \mathrm{DWConv1} \times 3\big(\mathrm{DWConv3} \times 1(T)\big) \tag{11}$$

$$T^{(0)} = \mathrm{DW}\big(X\big) \tag{12}$$

$$T^{(l)} = \mathrm{DW}\big(\mathrm{WT}(T^{(l-1)})\big), \quad l = 1, 2, 3 \tag{13}$$

where, $X$ denotes the input features, and $WT$ denotes the Haar wavelet transform. Specifically, when processing feature in wavelet domain, a parameter sharing strategy is adopted for better efficiency.

The Haar wavelet transform decomposes a feature into four components: LL, HL, LH, and HH. We define LL as the low-frequency component $T_L$ and group HL, LH, and HH as the high-frequency component $T_H$.

$$T_L^{(l)}, T_H^{(l)} = T^{(l)} \tag{14}$$

Leveraging the linearity of the wavelet transform, we process the wavelet components independently before being recomposed for the inverse transform. This process yields the aggregated wavelet feature $R^{(1)}$.

$$R^{(3)} = \mathrm{IWT}\big(T^{(3)}\big), \tag{15}$$

$$R^{(2)} = \mathrm{IWT}\big(T_L^{(2)} + R^{(3)}, T_H^{(2)}\big), \tag{16}$$

$$R^{(1)} = \mathrm{IWT}\big(T_L^{(1)} + R^{(2)}, T_H^{(3)}\big), \tag{17}$$

First, the aggregated wavelet feature $R^{(1)}$ is summed with the $T^{(0)}$. This combined feature, capturing diverse frequency characteristics, is then processed sequentially by a GELU activation and a $1 \times 1$ convolution for fusion. The final output is the wavelet modulation map $x_{map}$.

$$x_{map} = \mathrm{GELU}\big(\mathrm{Conv1x1}\big(\mathrm{GELU}(R^{(1)} + T^{(0)})\big)\big). \tag{18}$$

## C  TRAINING DETAILS AND HYPERPARAMTERS

All experiments used PyTorch. The detailed software and hardware configurations are listed below:

**Software Environment:**

- Operating System: Ubuntu 20.04
- Framework: PyTorch 2.4
- Libraries: CUDA 12.4, cuDNN 8.9

**Hardware Platform:**

- GPU: 1x NVIDIA GeForce RTX 4090 (24 GB VRAM)
- CPU: Intel(R) Xeon(R) Silver 4310 @ 2.10 GHz
- System Memory: 256 GB RAM

Table 8 shows all the hyperparameters used in experiment.

## D  DETAILS OF ARCHITECTURE FOR DIFFERENT BASE CODECS

In Figures 7, 8, and 9, we provide the detailed network architectures of the proposed Combine-ICMH framework for different image codecs. The Combine-ICMH framework consistently outperforms previous methods varying in architecture, all while preserving the pretrained model's R-D performance.

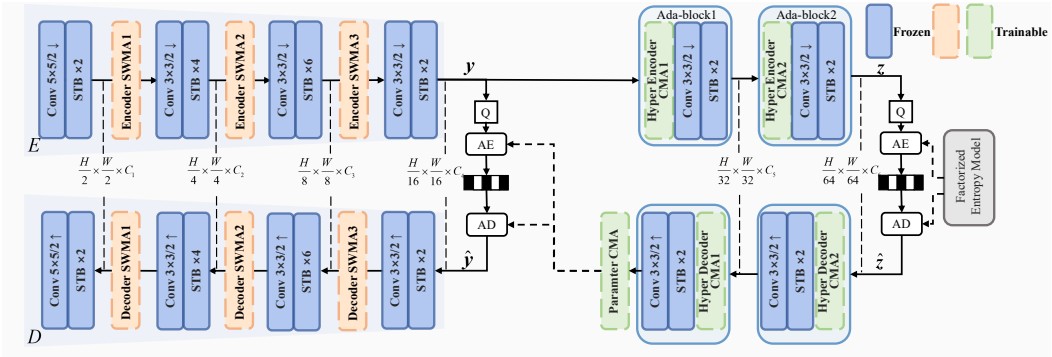

Figure 7: For a fair comparison, we follow the (Chen et al., 2023) and adopt a simplified version of the TIC model (Lu et al., 2021) as our base codec. In this model, STB denotes the Swin-Transformer Block (Liu et al., 2021).

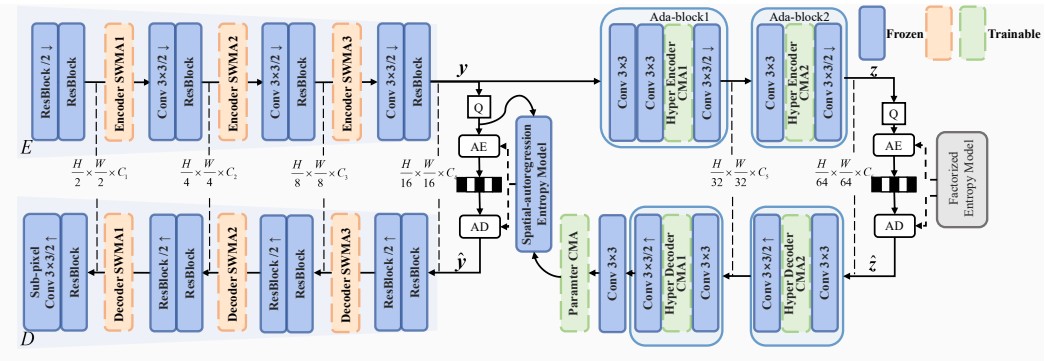

Figure 8: For further comparison, we use Cheng2020-anchor model Cheng et al. (2020) as base codec. In this model, Resblock denotes the residual block.

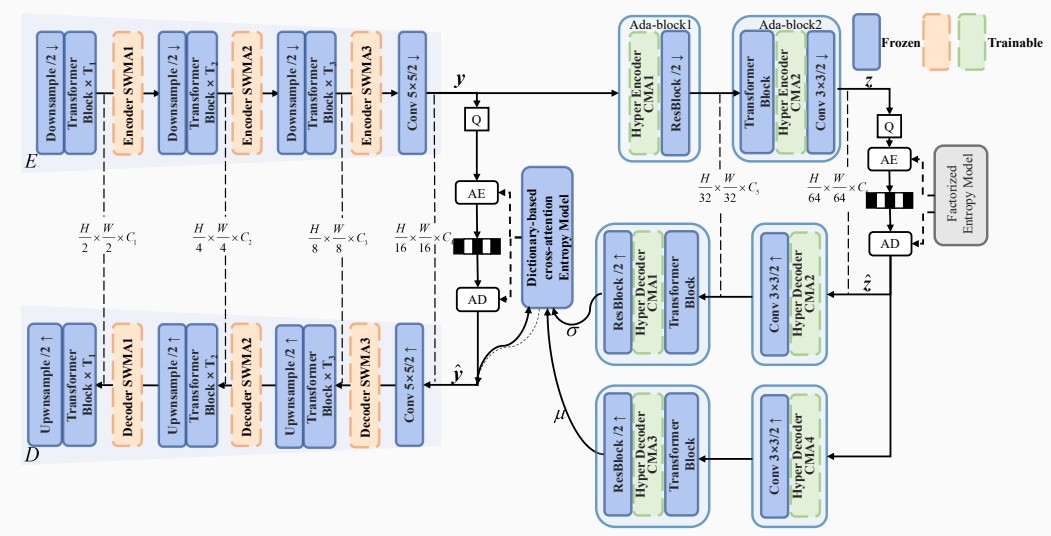

Figure 9: For further comparison, we use DCAE model Lu et al. (2025) as base codec. In this model, $Downsample$ denotes three stacked residual blocks and a stride convolution, and $Upsample$ denotes three stacked residual blocks and sub-pixel upsampling on the last convolution.

Table 8: Training hyperparamters for experiments.

|  | Classification | Detection | Segmentation |
|---|---|---|---|
| Optimizer | Adam | Adam | Adam |
| Batch size | 16 | 8 | 8 |
| Trade-off term $\lambda$ | [2.5, 3.5, 5, 6.7, 13] | [0.5, 0.875, 1.75, 3] | [0.35, 0.5, 0.875, 1.75, 3] |
| Epochs | 8 | 40 | 40 |
| Learning rate schedule | MultiStepLR | - | - |
| Milestones | [5,7] | - | - |
| Learning rate decay | 0.5 | - | - |
| Base learning rate | 1e-4 | 1e-4 | 1e-4 |

# E   ANALYSIS OF CHANNEL MODULATION

In a standard Learned Image Compression (LIC) framework, the non-linear transform $(g_a, g_s)$ and the entropy model $(h_a, h_s)$ are jointly optimized. The transform converts an input signal into a compact latent representation $\mathbf{y}$, while the entropy model estimates the probability distribution of its quantized version, $\hat{\mathbf{y}}$, to minimize coding redundancy.

In hyperprior-based models (Ballé et al., 2018), $\bar{\mathbf{y}} \triangleq (\mathbf{y} - \boldsymbol{\mu})/\boldsymbol{\sigma}$ is modeled as a standard spherical normal vector, Gaussianizing the source distribution to minimize coding overhead (Zhu et al., 2022). The parameters $(\boldsymbol{\mu}, \boldsymbol{\sigma})$ are generated by the hyperprior decoder $h_s$ from the hyper-latents $\hat{\mathbf{z}}$, which are in turn produced by the hyperprior encoder $h_a$:

$$\mathbf{z} = h_a(\mathbf{y}; \boldsymbol{\theta}_{h_a}) \tag{19}$$
$$\hat{\mathbf{z}} = Q(\mathbf{z}) \tag{20}$$
$$(\boldsymbol{\mu}, \boldsymbol{\sigma}) = h_s(\hat{\mathbf{z}}; \boldsymbol{\theta}_{h_s}) \tag{21}$$

where $Q(\cdot)$ denotes quantization.

In practice, the hyperprior-based LIC framework employs the Gaussian parameters $(\boldsymbol{\mu}, \boldsymbol{\sigma})$ to model the probability distribution of the quantized latent representation $\hat{\mathbf{y}}$ and calculate the bitrate $R(\hat{\mathbf{y}})$ (Feng et al., 2025):

$$R(\hat{\mathbf{y}}) = \mathbb{E}\left[-\log_2(p_{\hat{\mathbf{y}}|\hat{\mathbf{z}}}(\hat{\mathbf{y}}|\hat{\mathbf{z}}))\right] \tag{22}$$
$$p_{\hat{\mathbf{y}}|\hat{\mathbf{z}}}(\hat{\mathbf{y}}|\hat{\mathbf{z}}) \sim \mathcal{N}(\boldsymbol{\mu}, \boldsymbol{\sigma}^2) \tag{23}$$

However, during fine-tuning in previous frameworks (Chen et al., 2023; Li et al., 2024), only the transform modules $(g_a, g_s)$ are trained, while the entropy model $(h_a, h_s)$ is kept frozen. This introduces a "transform-entropy mismatch." Specifically, fine-tuning shifts the original latent $\mathbf{y}$ to a new task-specific latent $\mathbf{y}'$, altering its statistical distribution.

Consequently, the Gaussian parameters $(\boldsymbol{\mu}', \boldsymbol{\sigma}')$ generated by the frozen entropy model for the new latent $\mathbf{y}'$ are suboptimal. They are derived via the frozen hyperprior network:

$$\mathbf{z}' = h_a(\mathbf{y}'; \boldsymbol{\theta}_{h_a}) \tag{24}$$
$$\hat{\mathbf{z}}' = Q(\mathbf{z}') \tag{25}$$
$$(\boldsymbol{\mu}', \boldsymbol{\sigma}') = h_s(\hat{\mathbf{z}}'; \boldsymbol{\theta}_{h_s}) \tag{26}$$

This suboptimality manifests as a deviation $(\Delta\boldsymbol{\mu}, \Delta\boldsymbol{\sigma})$ from the ideal parameters $(\boldsymbol{\mu}_{ideal}, \boldsymbol{\sigma}_{ideal})$ that model $\mathbf{y}'$ into a standard spherical normal vector. The deviation is defined as:

$$\Delta\boldsymbol{\mu} = \boldsymbol{\mu}' - \boldsymbol{\mu}_{ideal} \tag{27}$$
$$\Delta\boldsymbol{\sigma} = \boldsymbol{\sigma}' - \boldsymbol{\sigma}_{ideal} \tag{28}$$

As a result, the normalized latent $\bar{\mathbf{y}}'$ deviates from the standard Gaussian distribution. This mismatch results in encoding redundancy. We can express the actual bitrate $R(\hat{\mathbf{y}}')$ by decomposing it into an ideal bitrate and a penalty term, $\Delta bpp$, which is defined as:

$$R(\hat{\mathbf{y}}') = \mathbb{E}\left[-\log_2(p_{\hat{\mathbf{y}}'|\hat{\mathbf{z}}'}(\hat{\mathbf{y}}' \mid \hat{\mathbf{z}}'))\right] = \mathbb{E}\left[-\log_2(p_{\hat{\mathbf{y}}'|\tilde{\mathbf{z}}}(\hat{\mathbf{y}}' \mid \tilde{\mathbf{z}}))\right] + \Delta bpp \tag{29}$$
$$p_{\hat{\mathbf{y}}'|\tilde{\mathbf{z}}}(\hat{\mathbf{y}}'|\tilde{\mathbf{z}}) \sim \mathcal{N}(\boldsymbol{\mu}_{ideal}, \boldsymbol{\sigma}_{ideal}^2) \tag{30}$$

where $\tilde{\mathbf{z}}$ is the assumed hyper-latent that enables the latent representation $\mathbf{y}'$ to be modeled as a standard Gaussian distribution, and $\Delta bpp$ represents the increased bpp compared to the ideal scenario where the entropy model and transform module are perfectly matched.

To mitigate the mismatch, we propose the Channel-Modulation Adapter (CMA) module, which is specifically designed to resolve the "transform-entropy mismatch" by adapting the hyperprior to meet the fine-tuned transform, thereby minimizing coding redundancy.

The proposed CMA mitigates the "transform-entropy mismatch" by adjusting the entropy model $(h_a, h_s)$. This adaptation is designed to minimize $\Delta\boldsymbol{\mu}$ and $\Delta\boldsymbol{\sigma}$. Ideally, if $\Delta\boldsymbol{\mu}$ and $\Delta\boldsymbol{\sigma}$ are driven to zero, the coding redundancy introduced by the fine-tuning transform module is completely eliminated.

In the ideal case:

$$R(\hat{\mathbf{y}}') = \mathbb{E}\left[-\log_2(p_{\hat{\mathbf{y}}'|\hat{\mathbf{z}}'}(\hat{\mathbf{y}}'|\hat{\mathbf{z}}'))\right] = \mathbb{E}\left[-\log_2(p_{\hat{\mathbf{y}}'|\tilde{\mathbf{z}}}(\hat{\mathbf{y}}'|\tilde{\mathbf{z}}))\right] \tag{31}$$

$$\Delta bpp = 0 \tag{32}$$

To further quantitatively analyze the effectiveness of the proposed CMA in reducing coding redundancy, we measure various correlations within the latent representation $y$, following the Zhu et al. (2022).

We compare two fine-tuning strategies at the lowest-bitrate setting, where redundancy is most pronounced: (i) a baseline that inserts only the SWMA adapter and (ii) the full approach that jointly optimizes SWMA and CMA. Both variants fine-tune the TIC codec (Lu et al., 2021) for object detection on the COCO2017-val dataset.

First, we estimate the inter-channel correlation of the latent representation $y$ by calculating the pairwise channel similarity, as depicted in Figure 10.

As shown in Figure 10, the baseline model (SWMA-only) exhibits high inter-channel correlation, implying significant redundancy among channels. In contrast, the model equipped with our CMA (SWMA+CMA) demonstrates a marked reduction in correlation, evidenced by a much steeper decay in the similarity curve. These findings strongly suggest that the CMA promotes the learning of more distinct, decorrelated features, thereby effectively reducing coding redundancy.

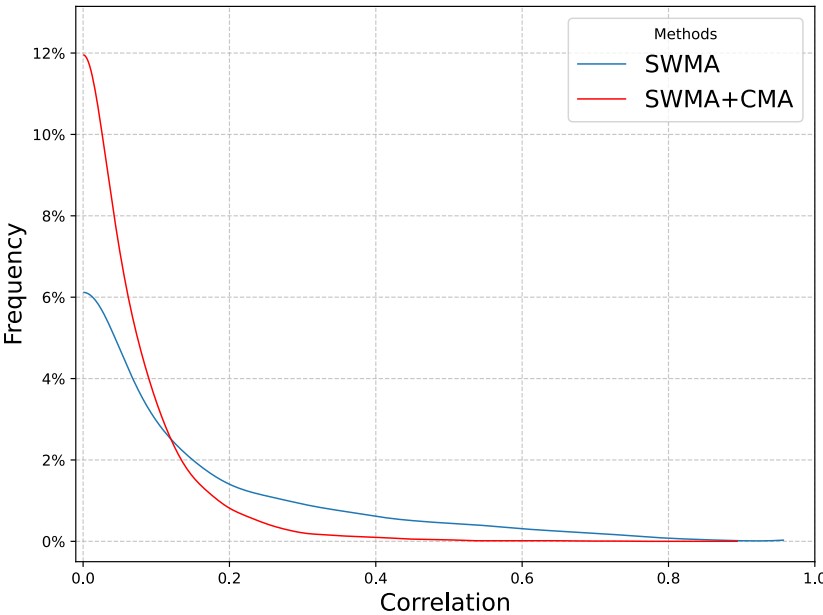

Figure 10: Pairwise channel similarities of the latent representation $y$ for different fine-tuning strategies. Incorporating CMA markedly reduce the inter-channel correlation.

Next, we extend our analysis to the spatial dimension. Following the Zhu et al. (2022), we measure the correlation between nearby spatial positions, which is averaged across all channels. As visualized in Figure 11, the results show a marked improvement. While the SWMA-only baseline exhibits a high average spatial correlation ($\rho$=0.3283), our SWMA+CMA model substantially reduces this to $\rho$=0.2439. This reduction in spatial redundancy directly corresponds to the improved downstream task performance, validating that our framework yields a more compact and effective latent representation.

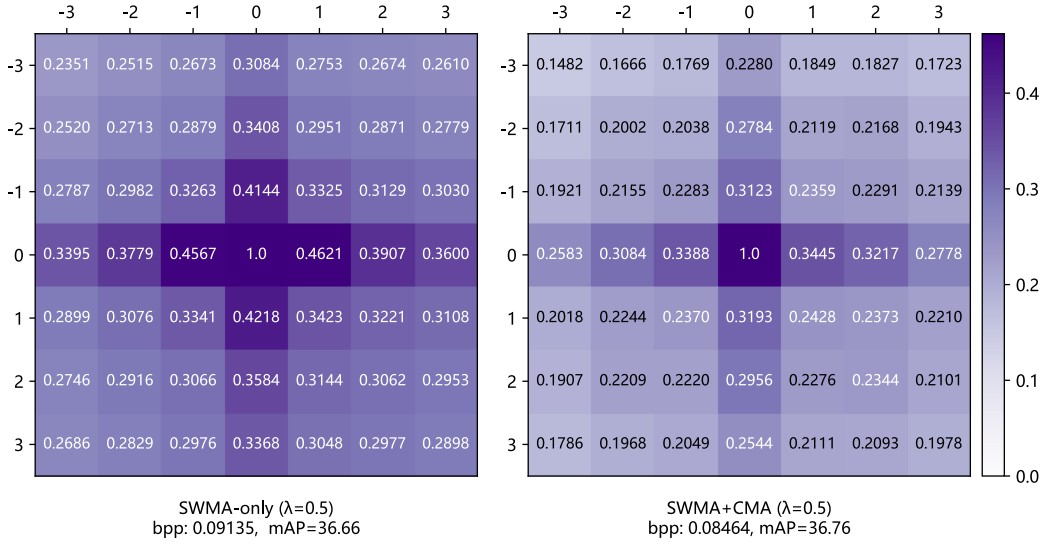

Figure 11: Spatial correlation of $(y - \mu)/\sigma$ with models trained at $\lambda = 0.5$. SWMA+CMA (right) achieves average smaller correlation than SWMA-only (left).

In summary, our comprehensive correlation analysis across both channel and spatial dimensions provides strong evidence for the effectiveness of our proposed fine-tuning paradigm. Prior methods Li et al. (2024); Chen et al. (2023), which only fine-tune the transform module, inevitably cause a "transform-entropy mismatch." Our approach directly addresses this issue by introducing the CMA module, which facilitates a collaborative optimization between the entropy model and the transform. The significant reduction in inter-channel and spatial correlation directly reflects an improvement in encoding efficiency, underscoring the benefit of this collaborative optimization.

Furthermore, we analyze the CMA's specific impact on the hyperprior. As illustrated in Figure 3, our approach differs from conventional channel attention mechanisms that multiplicatively scale each channel. Instead, the CMA computes an additive correction based on the input features $x$ to modify their distribution. Specifically, in a hyperprior-based entropy model, the CMA module directly operates on the output of the hyperprior network ($h_a$, $h_s$). This allows it to adapt the distribution estimation by modifying the Gaussian parameters ($\boldsymbol{\mu}, \boldsymbol{\sigma}$).

To evaluate this adaptation, we analyze the quantization loss of the latent representation $y$, following the methodology of (Xie et al., 2021b). The quantization loss during compression is quantified using a "scaling deviation" metric (Xie et al., 2021b), which is defined as $\epsilon = \text{abs}\left(\hat{\boldsymbol{y}} - \boldsymbol{y}\right)/\Sigma\boldsymbol{y}$. In the context of the Gaussian entropy model, the quantization process of the latent representation $y$ can be formulated as $\hat{\boldsymbol{y}} = Q(\boldsymbol{y} - \boldsymbol{\mu}) + \boldsymbol{\mu}$, so there is the following equation:

$$\epsilon = \text{abs}\left(Q(\boldsymbol{y} - \boldsymbol{\mu}) - (\boldsymbol{y} - \boldsymbol{\mu})\right)/\Sigma\boldsymbol{y} \tag{33}$$

Equation 33 reveals a direct link between the Gaussian parameter $\boldsymbol{\mu}$ and the quantization loss. A smaller distance between $\boldsymbol{\mu}$ and $\boldsymbol{y}$ results in lower quantization loss. Therefore, we use this metric as a proxy to evaluate our paradigm's effectiveness in adapting the hyperprior statistics.

Figure 12 presents the scaled deviation map on the COCO2017-val dataset. The results clearly demonstrate that our CMA module significantly reduces the quantization loss, particularly in background regions. This indicates that our paradigm, by adapting the Gaussian parameter $\boldsymbol{\mu}$, more

accurately estimates the statistical distribution of the shifted latent representation $\boldsymbol{y}'$, thereby mitigating the "transform-entropy mismatch".

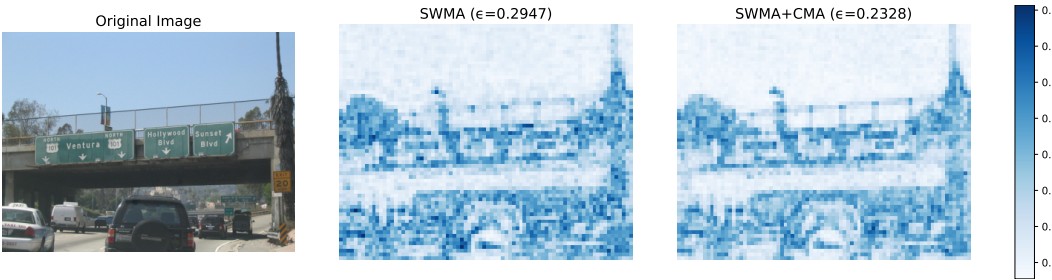

Figure 12: Scaled deviation map of two strategies.

Furthermore, we provide a qualitative analysis by visualizing the normalized latent representations $y$ from both fine-tuning strategies. As shown in Figure 13, which displays the eight highest-energy channels, the latent representation from the SWMA+CMA strategy exhibits significantly less discernible structure. This observation visually confirms the superior decorrelation capabilities of our fine-tuning paradigm.

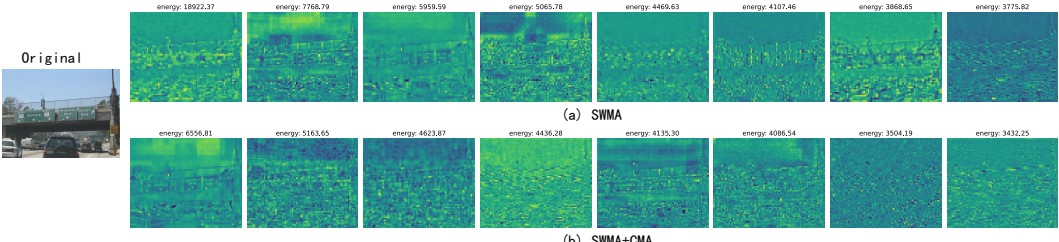

Figure 13: Each row corresponds to a different strategy and shows the eight channels with the highest entropy.

The proposed CMA module adapts the Gaussian parameters of the main entropy model by modulating the hyper-latent representation $z$ via the hyperprior network. As this hyperprior architecture is a foundational component in many state-of-the-art compression frameworks (Lu et al., 2025; Feng et al., 2025), our paradigm is broadly applicable. The potential of this approach is particularly evident in sophisticated models like auto-regressive entropy model (Minnen et al., 2018), where the conditional probability is formulated as:

$$p_y(\hat{y} \mid \hat{z}, \boldsymbol{\theta}_{h_s}, \boldsymbol{\theta}_{\mathrm{cm}}, \boldsymbol{\theta}_{\mathrm{ep}}) = \prod_i \left( \mathcal{N}(\boldsymbol{\mu}_i, \boldsymbol{\sigma}_i^2) * \mathcal{U}(-\frac{1}{2}, \frac{1}{2}) \right)(\hat{y}_i) \tag{34}$$

$$\text{with } \mu_i, \sigma_i = g_{\mathrm{ep}}(\psi, \boldsymbol{\phi}_i; \boldsymbol{\theta}_{\mathrm{ep}}), \psi = h_s(\hat{z}, \boldsymbol{\theta}_{h_s}), \text{and } \boldsymbol{\phi}_i = g_{\mathrm{cm}}(\hat{y}_{<i}, \boldsymbol{\theta}_{\mathrm{cm}}). \tag{35}$$

In such models, the Gaussian parameters are determined by the hyperprior ($\psi$), a context model ($g_{cm}$), and an entropy parameter network ($g_{ep}$). Our current work demonstrates that adapting just the core hyperprior component yields significant performance gains. This success establishes a promising new direction. Designing specialized adapters for the other components ($g_{cm}, g_{ep}$) is a valuable avenue for future research, potentially leading to even greater performance enhancements.

## F   MORE RESULTS OF THE CMA

Recent advancements in Image Compression for Machine and Human Vision (ICMH) have largely focused on fine-tuning the analysis and synthesis transforms (i.e., the encoder and decoder). For instance, TransTIC (Chen et al., 2023) leverages prompt-tuning by inserting modules into the Transformer's attention layer. In another line of work, Adapt-ICMH (Li et al., 2024) introduces a lightweight Space-Frequency Modulation Adapter (SFMA) to efficiently fine-tune the encoder

and decoder for reducing spatial and frequency redundancy. Another approach (Park et al., 2025) has utilized SVD-LoRA to fine-tune the codec, combined with Test-Time Fine-tuning (TTFT) for instance-specific optimization. A common thread in these methods is their exclusive focus on adapting the transform modules, neglecting the entropy model.

In contrast, our work stems from a key observation: neglecting the entropy model during fine-tuning leads to a "transform-entropy mismatch." This mismatch increases coding redundancy, which degrades both coding efficiency and overall task performance. To address this, we propose the Channel Modulation Adapter (CMA), a lightweight adapter designed specifically to adapt the entropy model and resolve this mismatch.

To validate our claim, we compared four architectural configurations: SWMA-only, SWMA + CMA, SFMA-only, and SFMA + CMA. As illustrated in Figures 14 and 15, integrating our CMA yields a significant reduction in bit-per-pixel (bpp) without compromising task performance. This result confirms that CMA effectively mitigates coding redundancy in the ICMH fine-tuning process.

The quantitative results in Table 9 further corroborate this finding. Specifically, the CMA provides an average bitrate saving of 5.05% when integrated with the SWMA baseline and 3.89% with the SFMA baseline. Its effectiveness across different models further highlights the importance of fine-tuning the entropy model in the ICMH.

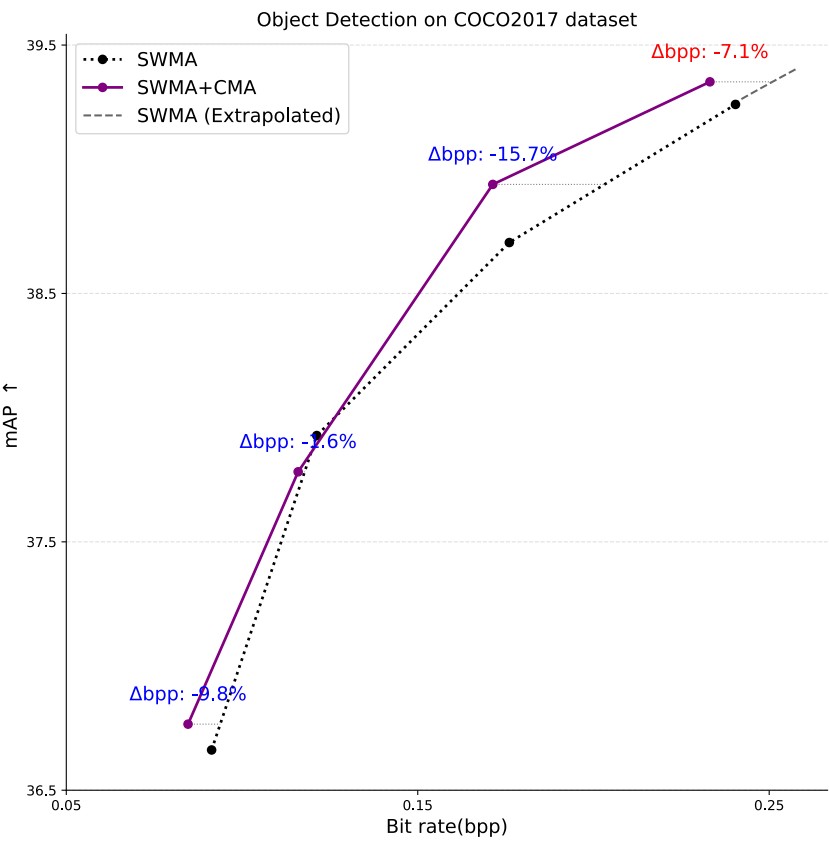

Figure 14: Object-detection performance on the COCO2017-val dataset with TIC (Lu et al., 2021) as the basic codec. Incorporating CMA markedly reduces the bitrate without compromising detection accuracy.

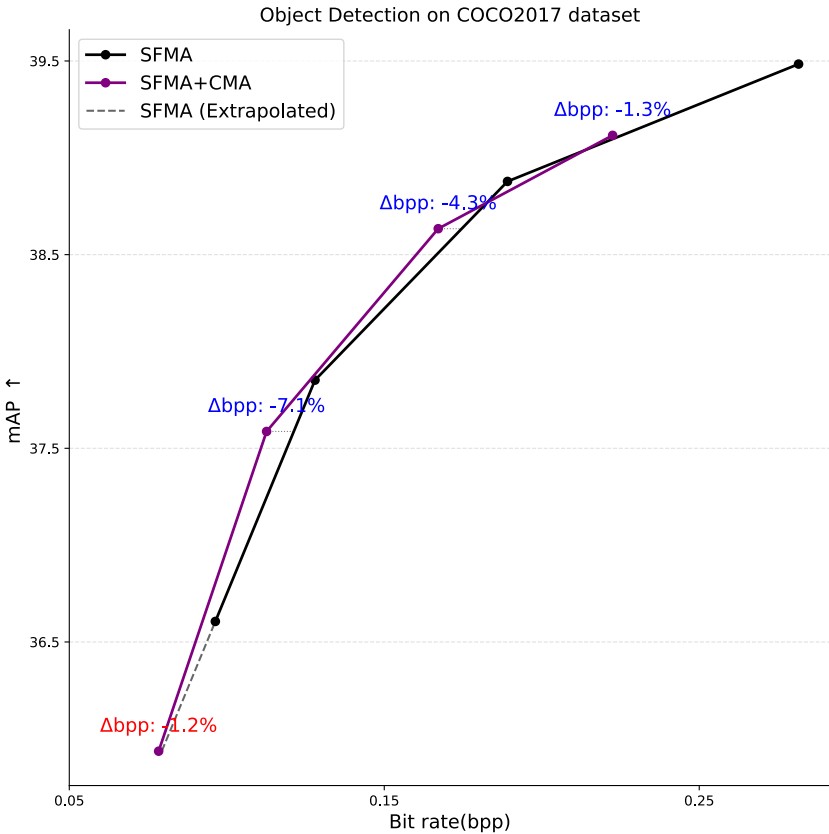

Figure 15: Object-detection performance on the COCO-2017 dataset with TIC (Lu et al., 2021) as the basic codec. Incorporating CMA markedly reduces the bitrate without compromising detection accuracy.

Table 9: Performance comparison of different methods on Object Detection, using TIC as the base codec. We report the number of trainable parameters and two BD metrics (Bjontegaard, 2001): BD-rate and BD-acc/mAP.

| Method | Detection | | Trainable |
|---|---|---|---|
| | BD-rate↓ | BD-mAP↑ | Params ↓(M) |
| SWMA-only | 0.00% | 0.00 | 0.276 (3.7%) |
| SWMA+CMA | -5.05% | 0.17 | 0.319 (4.2%) |
| SFMA-only | 0.00% | 0.00 | 0.287 (3.8%) |
| SFMA+CMA | -3.89% | 0.10 | 0.331 (4.4%) |

# G    MORE RESULTS

## G.1    IMPLEMENTATION ON CNN-BASED IMAGE CODECS

To further validate the generalizability of our method, we conduct supplementary experiments on cheng20-anchor (Cheng et al., 2020), a CNN-based model featuring an auto-regressive entropy model. Its CNN-based architecture and moderate complexity (26.6M) provide a sharp contrast to the low-complexity (TIC (Lu et al., 2021)) and high-complexity (DCAE Lu et al. (2025)) Transformer-based models from our main experiments, thus offering a rigorous test for our paradigm's adaptability. As shown in Fig. 16, Fig. 17 and Table 10, on the object detection task and instance segmentation task, our combine-ICMH again achieves superior performance with a comparable parameter count to the baseline ICMH framework. This result provides strong evidence for the robustness and adaptability of our proposed method across different architectures and entropy models.

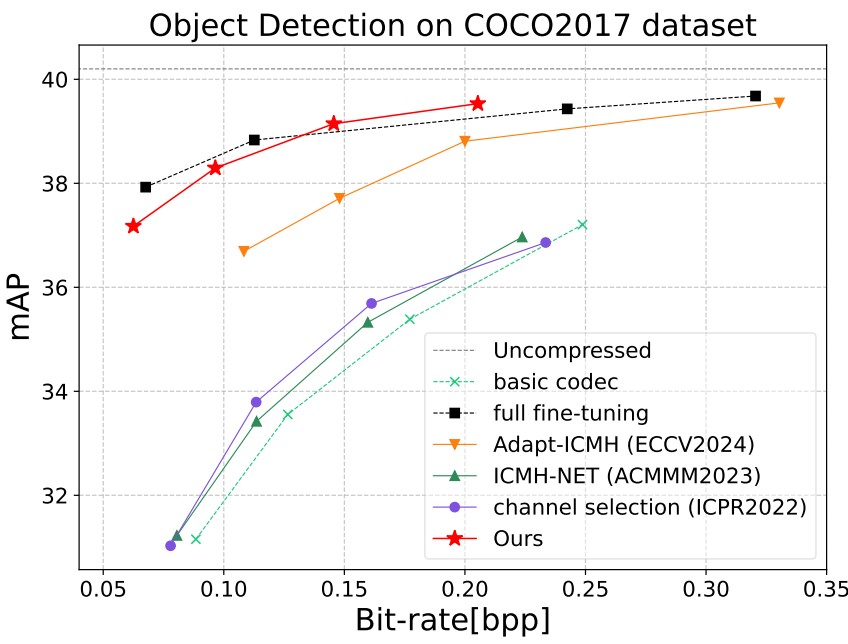

Figure 16: Performance comparison for object detection on the COCO2017 dataset using the cheng2020-anchor.

Table 10: Performance comparison for object detection on the COCO2017 dataset using the cheng2020-anchor.

| Method | Detection | | Segmentation | | Trainable |
|---|---|---|---|---|---|
| | BD-rate↓ | BD-mAP↑ | BD-rate↓ | BD-mAP↑ | Params ↓(M) |
| full fine-tuning | -61.65% | 4.69 | -73.45% | 3.86 | 26.60 (100%) |
| ICMH-NET | -8.31% | 0.50 | -11.67% | 0.71 | 4.43 (16.6%) |
| channel selection | -11.66% | 0.73 | -5.16% | 0.23 | 1.34 (4.8%) |
| Adapt-ICMH | -49.21% | 3.12 | -60.71% | 3.40 | 0.41 (1.5%) |
| Ours | **-61.45%** | **4.40** | **-65.03%** | **3.55** | 0.43 (1.6%) |

## G.2    ABLATION STUDIY ON THE POSITION OF THE CMA

We conducted an ablation study on the COCO2017-val dataset to determine the optimal placement of the CMA module for the object detection task. Our baseline is the Adapt-ICMH framework (at its first quality level) with the CMA module disabled. We integrated the module at three distinct locations within the hyperprior network— within the hyper-encoder ($h_a$), within the hyper-decoder ($h_s$), and after the entire network—and evaluated the resulting mAP gain under matched bpp conditions.

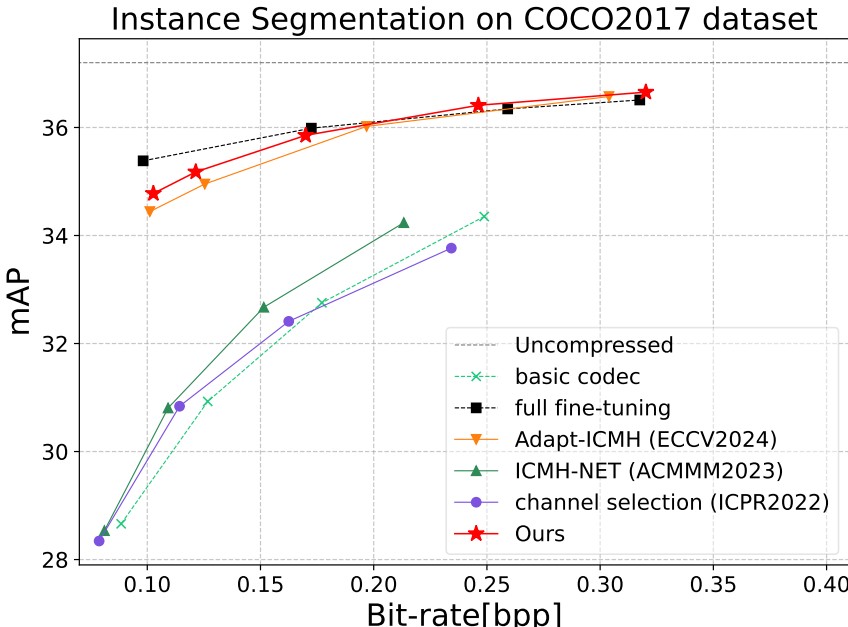

Figure 17: Performance comparison for instance segmentation on the COCO2017 dataset using the cheng2020-anchor.

The results, presented in Table 11. First, they validate the overall effectiveness of the CMA module, as all configurations significantly outperform the baseline. Further comparison indicates that the optimal strategy for maximum performance gain is to place the module after the hyper-prior's feature extraction module (corresponding to configurations b and e). Interestingly, placing the CMA after the entire hyperprior network also yields substantial improvements. We attribute this enhancement to the module's ability to directly make adjustments to the final output features of the hyperprior entropy model.

Table 11: Ablations on middle dimension of CMA. Performance comparison for object detection on the COCO2017 dataset.

|     | $h_a$ | $h_s$ | after $h_s$ | $gain$ | Param(M) |
|-----|-------|-------|-------------|--------|----------|
| (a) | \     | \     | \           | 0.00   | 0.287(3.8%) |
| (b) | 1,3   | 0,2   | \           | 0.423  | 0.305(4.1%) |
| (c) | 1,2,3 | 0,1,2 | \           | 0.340  | 0.319(4.2%) |
| (d) | 1,3   | 1,3   | ✓           | 0.545  | 0.358(4.8%) |
| (e) | 1,3   | 0,2   | ✓           | 0.713  | 0.343(4.6%) |

### G.3 ABLATION STUDIY ON THE MIDDLE DIMENSION OF THE CMA

To investigate the optimal middle dimension of CMA, we conduct an ablation study on the downstream task of object detection using the COCO2017-val dataset. Our baseline is the Adapt-ICMH framework (at its first quality level) with the CMA module disabled. We then integrate CMA variants with different middle dimensions at the $h_a$, $h_s$ and evaluate the mAP gain for each configuration under matched bpp conditions.

Table 12 reveals a clear trade-off in the CMA's middle dimension. Increasing the dimension to 16 substantially boosts performance, but further increases yield negligible gains while the parameter count grows steadily. Thus, we select 16 as the optimal dimension, balancing performance with model complexity.

Notably, a sharp degradation in performance is observed when the middle dimension is increased to 64. We hypothesize that this is due to over-parameterization, where an excessively complex adapter may hinder the fine-tuning process or introduce training instability.

Table 12: Ablations on middle dimension of CMA. Performance comparison for object detection on the COCO2017 dataset.

|     | middle dimension | $gain$ | Param(M) |
| --- | --- | --- | --- |
| (a) | 1 | 0.046 | 0.289(3.8%) |
| (b) | 16 | 0.423 | 0.305(4.1%) |
| (c) | 32 | 0.354 | 0.324(4.3%) |
| (d) | 64 | -0.047 | 0.358(4.8%) |

### G.4 ABLATION STUDIY ON THE VARIANT OF THE CMA

We conduct a comparative study of our $GAP \to Conv1 \times 1 \to SiLU$ operation within the CMA module against two attention mechanisms: ECA (Wang et al., 2020), and MCA (Jiang et al., 2024). The experiment was performed on the COCO2017-val dataset. For a fair comparison, all variants used the same optimal placement for the CMA (configuration (b) from Table 11), and their performance was measured as the mAP gain over the Adapt-ICMH baseline. As shown in Table 13, the results confirm that our proposed operation decisively outperforms both alternatives, highlighting its superior design for this task.

Table 13: Ablations on middle dimension of CMA. Performance comparison for object detection on the COCO2017 dataset.

| Method | $gain$ | Param(M) |
| --- | --- | --- |
| Ours | 0.423 | 0.305(4.1%) |
| ECA | 0.073 | 0.304(4.1%) |
| MCA | 0.269 | 0.305(4.1%) |

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

## H COMPARISON TO OTHER RELATED WORK

While our analysis focuses on single-task baselines, we notice current work in multi-task adaptation, such as the multi-path aggregation scheme from Zhang et al. (2024) and the multi task transfer techniques from Zhao et al. (2025). Our study, however, follows the single-task, adapter-based paradigm (Chen et al., 2023; Li et al., 2024; Park et al., 2025). A direct quantitative comparison is not feasible due to the specialized, multi-task loss used in the other works. Nonetheless, adapting our analysis for multi-task scenarios remains a valuable avenue for future research.

## I THE USE OF LARGE LANGUAGE MODELS

A publicly available large-language model was used exclusively for minor grammar and wording corrections after the full manuscript—including all technical content, proofs, and equations—had been drafted by the authors. No text, figures, tables, or results were generated by the model. The authors manually verified every suggestion and assume full responsibility for the final content. The LLM is not listed as an author.

