# OpenReview forum: "Combine-ICMH: A Dual-Adapter Co-Tuning Framework in Image Compression for Machine and Human Vision"
_ICLR.cc/2026/Conference — ICLR 2026 Conference Withdrawn Submission_

### Official Review · Reviewer_gyCv · 2025-10-21

**Soundness:** 2
**Presentation:** 3
**Contribution:** 2
**Rating:** 2
**Confidence:** 4

**Summary:**

This paper aims to enhance parameter-efficient fine-tuning (PEFT) methods for Image Compression for Machine and Human Vision (ICMH). The authors argue that prior works, such as Adapt-ICMH, suffer from two key limitations: 1) inflexible frequency-domain adaptation and 2) a "transform-entropy mismatch" that arises from fine-tuning only the transform network while keeping the entropy model frozen. To address this, the paper introduces Combine-ICMH, a framework featuring two distinct adapters: a wavelet-based Spatial-Wavelet Modulation Adapter (SWMA) for the transform network and a Channel Modulation Adapter (CMA) for the entropy model. The core idea is to enable the synergistic co-tuning of both modules to improve rate-distortion-task performance.

**Strengths:**

1.The paper is well-written, clearly structured, and easy to follow. The authors effectively articulate the problem they aim to solve, the details of their proposed method, and the interpretation of their experimental results, which facilitates reader comprehension.

2.The authors provide solid empirical validation through extensive experiments. The method is shown to outperform multiple state-of-the-art baselines, and its components are well-justified through detailed ablation studies and generalization tests, demonstrating consistent performance gains.

**Weaknesses:**

1.The paper's efficiency evaluation is incomplete, it relies on trainable parameter counts and kMACs but neglects to report crucial inference latency metrics. The use of operations like DWT/iDWT in the SWMA, which are parameter-free yet computationally demanding, makes this omission problematic.

2.The paper's primary limitation is its lack of architectural novelty, with the proposed solutions appearing more as an engineering refinement of a pre-existing method  (i.e., Adapt-ICMH) than the introduction of a new paradigm. The CMA module is  a standard channel attention mechanism, and its application to the entropy model, while effective, is a straightforward engineering choice. Similarly, the SWMA module's replacement of FFT with a wavelet transform leverages a common alternative in signal processing and does not represent a significant conceptual advance in adapter design

**Questions:**

I am confused by the results for "SMA (ours)" in Table 3. What is the difference compared to the baseline "SMA"？Why can it improve performance with a lower parameter count?

---

### Official Review · Reviewer_wnAU · 2025-10-28

**Soundness:** 2
**Presentation:** 3
**Contribution:** 2
**Rating:** 2
**Confidence:** 5

**Summary:**

This paper proposes Combine-ICMH, a tuning framework for adapting pre-trained image codecs to both machine and human vision tasks. It tries to solve "transform-entropy mismatch," as the frozen entropy model cannot adapt to the altered latent feature distribution.

**Strengths:**

1. The paper is well-organized and easy to follow.

2. The implementation details are well-presented, making the work easy to reproduce.

**Weaknesses:**

1. This work appears to be an incremental improvement over the Adapt-ICMH ramework. The primary contributions are the addition of an adapter for the entropy model (the CMA) and replacing the frequency-domain transform from FFT to a Wavelet transform (the WMA). Furthermore, the paper's core claim of addressing the "transform-entropy mismatch" is handled by applying the CMA to the hyperprior network ($h_a, h_s$). The analysis does not seem to extend to more complex entropy models. For a paper focused on resolving entropy mismatch, this limited scope seems to lack sufficient depth.

2. In the main comparison tables (Table 2 and Table 6), the proposed method consistently utilizes more trainable parameters than the Adapt-ICMH baseline. While the absolute increase is small, this difference is not negligible relative to the small size of the adapters themselves (e.g., 0.319M for "Ours" vs. 0.287M for "Adapt-ICMH" in Table 2). Despite this larger parameter budget, the corresponding performance gains appear relatively minor.

3. There appears to be a significant inconsistency in the results reported for the cheng2020-anchor baseline. In Figure 16 (Object Detection), the proposed method ("Ours") demonstrates a very large performance lead over Adapt-ICMH across all bit-rates. However, in Figure 17 (Instance Segmentation) on the exact same cheng2020-anchor baseline, the performance gap is drastically reduced, with the proposed method showing only a marginal improvement over Adapt-ICMH. The authors should explain the source of this discrepancy. Why would the relative performance gain differ so substantially between object detection (using Faster R-CNN) and instance segmentation (using Mask R-CNN) when run on the identical base codec?

**Questions:**

see weakness. My main concern is that the improvements over Adapt-ICMH appear minor. Furthermore, the paper does not explore the 'transform-entropy mismatch' problem in significant depth.

---

### Official Review · Reviewer_rRvp · 2025-10-29

**Soundness:** 2
**Presentation:** 2
**Contribution:** 2
**Rating:** 2
**Confidence:** 5

**Summary:**

Overall, this paper proposes an adapter-based fine-tuning approach for ICMH, with a clear extension idea and well-defined motivation. The introduction and empirical analysis identify the limitations of existing methods such as Adapt-ICMH. The methodological innovation mainly lies in the SWMA and CMA modules. Experiments demonstrate consistent improvements in BD-rate and BD-mAP across different tasks. However, the conceptual advancement over Adapt-ICMH is relatively limited, representing an incremental improvement rather than a substantive innovation. Moreover, the analysis of why the CMA can improve entropy alignment remains largely empirical, and the paper lacks theoretical validation.

**Strengths:**

1. The proposed architecture clearly delineates the placement of SWMA and CMA modules in the codec pipeline.
2. The Spatial–Wavelet Modulation Adapter (SWMA) shows notable advantages in both performance and parameter efficiency. The architecture consistently improves results on classification, detection, and segmentation tasks, delivering better overall rate–distortion and task performance with fewer additional parameters. Ablation studies further indicate complementary benefits from the Spatial Modulation Adapter (SMA) and the Wavelet Modulation Adapter (WMA): SMA enhances low-frequency context, while WMA preserves high-frequency details. Together they form a balanced and effective modulation mechanism that surpasses prior SFMA-based designs.
3. The CMA module leverages the hyperprior network to realign the adapted latent distributions, thereby directly calibrating the probability estimates of the entropy model.

**Weaknesses:**

1. The paper mainly builds on Adapt-ICMH[1] by replacing the prior SFMA module with the proposed SWMA, which yields performance gains but constitutes an incremental rather than a substantively novel contribution. Regarding entropy coding, the added CMA module to refine the latent-feature statistics is indeed new. However, the performance at high bitrates is not better than Adapt-ICMH. Moreover, the comparison with existing methods and the articulation of the underlying motivation are not sufficiently in depth.
2. The paper does not present any encoder/decoder runtime comparisons; it only analyzes the parameter counts of the fine-tuned models.
3. No visualization or divergence metric (e.g., KL-divergence) is presented to support the “distribution alignment” claim.
4. Some typos: Fig.1 "covert" maybe "convert"; Fig.2 "Frequncy" -> "Frequency".

[1]. Li, Han, et al. "Image compression for machine and human vision with spatial-frequency adaptation." European Conference on Computer Vision. Cham: Springer Nature Switzerland, 2024.

**Questions:**

1. To what extent does the paper build on Adapt-ICMH [1] rather than present a substantively novel contribution, given that the SWMA replaces the prior SFMA module and the CMA module refines the latent-feature statistics but still shows no improvement at high bitrates? Also, are the comparisons and motivations analyzed in sufficient depth?
2. How is the location of the CMA module determined? Why is it placed before the Hyper Encoder/Decoder modules?
3. What is the theoretical basis for the claim that wavelet transforms improve task adaptability?
4. Although the adapters in this paper introduce only a small number of trainable parameters, the overall complexity remains comparable to that of the full codec, and, in practice, multiple models often have to be maintained. This is inconvenient for deploying image compression networks on edge devices and further introduces model-switching overhead.
5. If the adaptation method proposed in this paper is applied to a more recent image compression model, such as MLIC++[2], would it still deliver comparable gains?

[2]. Jiang, Wei, et al. "MLIC++: Linear Complexity Multi-Reference Entropy Modeling for Learned Image Compression." ACM Transactions on Multimedia Computing, Communications and Applications 21.5 (2025): 1-25.

---

### Official Review · Reviewer_gBGV · 2025-11-01

**Soundness:** 2
**Presentation:** 3
**Contribution:** 2
**Rating:** 4
**Confidence:** 5

**Summary:**

This paper proposes Combine-ICMH, a fine-tuning framework for Image Compression for Machine and Human Vision (ICMH). It targets two limitations in existing adapter-based methods: limited frequency adaptability in transform networks and transform-entropy mismatch resulting from distribution shifts between adapted latents and the fixed entropy model. Combine-ICMH incorporates two lightweight adapters: the Spatial-Wavelet Modulation Adapter (SWMA) to enhance multi-scale frequency representation, and the Channel Modulation Adapter (CMA) to adjust the entropy model to the modified latent distribution. Experimental results show that Combine-ICMH achieves state-of-the-art performance on multiple machine vision tasks with low parameter cost.

**Strengths:**

1.The manuscript is clearly written and well-structured, with high-quality figures. Notably, the power spectral density (PSD) analysis in Figure 5 demonstrates that incorporating discrete wavelet transform (DWT) in the lightweight adapter preserves image information more effectively than alternatives.

2.The method is evaluated through comprehensive experiments, consistently outperforming several state-of-the-art baselines across multiple tasks.

**Weaknesses:**

1.The contributions appear largely incremental. The proposed Spatial-Wavelet Modulation Adapter (SWMA) is a modest extension of the Spatial-Frequency Modulation Adapter (SFMA) from Adapt-ICMH (Li et al., 2024), differing primarily in the substitution of FFT with DWT in the frequency branch. Similarly, the Channel Modulation Adapter (CMA) is a straightforward channel-wise adaptation mechanism for entropy models, with limited discussion or justification. The absence of analysis regarding entropy model characteristics leaves unclear whether this design is optimal or could be further refined.

2.The placement of CMA across different codec architectures, as shown in Figure 9, lacks systematic justification and appears ad hoc. This raises questions about whether reported gains stem from the adapter design itself or from architecture-specific tuning.

3.The claim of computational efficiency, supported only by kMACs/pixel, is unconvincing. This metric likely underestimates the cost of repeated DWT operations in SWMA, as their complexity is not fully captured. A fair comparison with FFT-based baselines requires actual encoding and decoding latency measurements, which are not provided.

4.The rate reduction estimation in Figures 14 and 15 (Appendix E) is methodologically flawed. Rate-mAP curves should be derived via polynomial interpolation (e.g., cubic, as in BD-rate/BD-PSNR calculations) of discrete operating points. Instead, the authors use linear interpolation and extrapolation, then compute reductions using points from the outer envelope and extrapolated segments—leading to potentially overestimated savings.

**Questions:**

1. CMA Design and Alternatives: The channel-wise adaptation of the entropy model receives little justification. What specific properties (e.g., conditional dependencies, hyperprior structure) motivated this approach? Were alternatives (e.g., scale/bias modulation, partial hyperprior fine-tuning) evaluated, and if so, why was channel-wise modulation selected?

2. SWMA Efficiency Gains: SWMA reportedly achieves superior performance with fewer parameters by replacing linear layers with 1×1 convolutions. What is the mechanism behind this improvement—enhanced representational capacity, better spatial adaptation, or another factor?

3. CMA Placement (Figure 9): CMA insertion points differ across codec architectures. What guided these choices? Was placement determined systematically or empirically per model? An ablation on placement consistency would clarify the generalizability of gains.

4. Computational Efficiency Metrics: Efficiency is assessed solely via kMACs/pixel, likely underestimating the cost of repeated DWT operations in SWMA. Why were encoding/decoding latencies on standard hardware omitted? Please provide wall-clock timings compared to FFT-based baselines for a complete efficiency evaluation.

---

### Note · Authors · 2025-11-12

I have read and agree with the venue's withdrawal policy on behalf of myself and my co-authors.